# Low Cycle Fatigue Behavior of Plastically Pre-Strained HSLA S355MC and S460MC Steels

**DOI:** 10.3390/ma15227927

**Published:** 2022-11-09

**Authors:** Christos G. Prosgolitis, Alexis T. Kermanidis, Helen Kamoutsi, Gregory N. Haidemenopoulos

**Affiliations:** 1Laboratory of Mechanics and Strength of Materials, Department of Mechanical Engineering, University of Thessaly, Pedion Areos, 38334 Volos, Greece; 2Laboratory of Materials, Department of Mechanical Engineering, University of Thessaly, Pedion Areos, 38334 Volos, Greece

**Keywords:** low cycle fatigue, plastic pre-straining, cyclic softening, fracture surface

## Abstract

Cold roll forming used in the manufacturing of lightweight steel profiles for racking storage systems is associated with localized, non-uniform plastic deformations in the corner sections of the profiles, which act as fatigue damage initiation sites. In order to obtain a clearer insight on the role of existing plastic deformation on material fatigue performance, the effect of plastic pre-straining on the low cycle fatigue behavior of S355MC and S460MC steels was investigated. The steels were plastically deformed at different pre-strain levels under tension, and subsequently subjected to cyclic strain-controlled testing. Plastic pre-straining was found to increase cyclic yield strength, decrease ductility, and induce cyclic softening, which, in S460MC, degrades fatigue resistance compared to the unstrained material. In unstrained conditions, the materials present a cyclic softening to hardening transition with increasing plastic strain amplitude, which in S355MC occurs at lower strain amplitudes and degrades its fatigue resistance with regard to the pre-strained material. Pre-straining also leads to a reduction in transition life from low to high cycle fatigue. SEM fractography, performed following the onset of crack initiation, revealed that plastic pre-straining reduces the fatigue fracture section as well as striation spacing, predominantly in the S355MC steel.

## 1. Introduction

Plastic deformation [1,2] induced by the cold forming process in the manufacturing of structural steel profiles is associated with work-hardening phenomena and the generation of non-uniform residual stresses [3] and strains [4] at specific locations [5,6,7], which, under alternating stresses, may influence the fatigue life of the component. The design of fatigue-resistant cold-formed sections requires a thorough understanding, at the material level, of the effect of existing plastic deformation on LCF behavior. It is known that the imposed plastic deformation develops microscopically [8,9] and creates a network of dislocations in the grains, dislocation pile-ups at the grain boundaries [10], and internal defects, which inhibit further dislocation movement and thus increase flow stress [11] and decrease ductility [12], which has been associated with inferior low cycle fatigue performance in steels [13,14].

The pre-straining effect on the LCF of steels has been examined partly under the viewpoint of its influence on fatigue life or cyclic hardening/softening behavior. Simultaneous correlations of LCF performance with hardening/softening mechanisms and conditions related to the onset and evolution of fatigue cracking are rarely reported. Research results have shown that the cyclic response of a plastically pre-strained material is dependent on material microstructure, the magnitude and direction of plastic pre-straining, and the applied cyclic plastic strain range. Das et al. [15] performed fully reversed strain-controlled fatigue tests on a DP600 dual-phase steel after uniaxial pre-straining of 12.5% in the rolling and transverse direction. They showed that, in reference conditions, the material exhibits mild cyclic hardening in the initial cycles followed by gradual cyclic softening until failure, whereas all pre-strained materials showed continuous cyclic softening. Sherman and Davies [16] found that, for the 0.11C–1.4Mn–0.48Si–0.08V DP steel with 18% martensite, tensile pre-straining increases fatigue life by about 50%, regardless of the magnitude of the pre-strain level (in the range of 2 to 8%). In Sherman et al.’s [17] study on hot-rolled low-carbon (HRLC) 0.05C–0.4Mn and high-strength low-alloy (HSLA) 0.14C–1.5Mn–0.6Si–0.04Al–0.16V–0.1N steels, it was revealed that, after 14% tensile pre-straining, fatigue life was slightly increased in the HRLC steel but decreased in the HSLA steel in comparison to the non-pre-strained specimens. The pre-strained materials showed cyclic softening behavior, regardless of the behavior in unstrained conditions; HRLC presented mixed-type (cyclic softening/ hardening) and HSLA presented cyclic hardening. Tensile pre-straining was found to be more detrimental to fatigue resistance than compressive pre-straining. Aichbhaumik [18] observed that, in the 17C–0.9Mn–0.85Si–0.12Zr HSLA steel, fatigue life was slightly decreased after 10% tensile pre-strain at high strain amplitudes but was increased at low strain amplitudes. The material exhibited cyclic softening behavior after plastic pre-straining, a shift from the mixed behavior in unstrained conditions. Raman and Padmanabhan [19] studied the effect of prior cold working on the low cycle fatigue behavior of AISI 304LN austenitic stainless steel, showing that the fatigue life increased at strain amplitudes below 0.5% while decreasing at higher strain amplitudes. The material presented cyclic softening behavior and the transition life decreased with increasing pre-strain level. Ghosal et al. [20] performed fully reversed strain-controlled tests to investigate the low cycle fatigue behavior of a DP590 steel after 10% uniaxial and equi-biaxial pre-straining. They reported that fatigue life was reduced in the pre-strained specimens, mainly after equi-biaxial pre-straining. Material in all pre-strained conditions revealed cyclic softening throughout the fatigue life. In other metallic alloys, the dependence of cyclic hardening/softening behavior on pre-straining has also been acknowledged. In a study by Jin et al. [21], it was shown that, in 304 stainless steel, a small initial hardening phase is followed by extensive cyclic softening due to tensile or torsional pre-straining and then secondary hardening associated with martensitic transformation. In commercial pure titanium, Chang et al. [22] found that cyclic softening is the dominant behavior of CP-Ti after pre-straining, and secondary cyclic hardening effects in opposition to the unstrained samples were not present. In the study by Singh et al. [23] on the LCF performance of 6061 and 2024 aluminum alloys, it was presented that, irrespective of the tensile pre-strain level (0, 4, 8%) examined, cyclic softening behavior was present in both alloys. Branco et al. [24] showed that the fatigue life of a pre-strained 7050-T6 aluminum alloy decreases as the tensile pre-strain level increases (from 0% to 8%) compared to an unstrained alloy. Similar to [23], the material displays cyclic strain-softening behavior, regardless of the level of pre-strain.

The current research focuses on the generation of new comparative experimental results regarding the LCF performance of plastically pre-strained S355MC and S460MC steels. The materials were subjected to plastic pre-stretching prior to cyclic testing, and the obtained LCF behavior was assessed under different levels of plastic pre-straining and compared to the behavior of initially unstrained materials. The experimental fatigue lives are combined with the cyclic hardening/softening characteristics of the materials and SEM fractography to investigate fracture characteristics following the onset of fatigue crack initiation.

## 2. Materials and Experimental Procedure

The materials used in the present investigation were the HSLA S355MC and S460MC steels in sheet form (as received from a coil), with a nominal thickness of 3.5 mm. The chemical composition of the materials is given in Table 1.

Monotonic plastic pre-straining under tension was applied on both materials, using the pre-straining process presented in Prosgolitis et al. [25], perpendicular to the rolling direction, so that the external cyclic loading is parallel to the pre-straining direction and transverse to the rolling direction. Low cycle fatigue (LCF) tests were performed on specimens with the geometry shown in Figure 1a in accordance with the SEP 1240 [26] guideline. Specimens were cut from the steel sheets using electro discharge machining (EDM). During testing, a 10 mm gauge length, dynamic, axial clip-on extensometer was used, and an anti-buckling device was attached to the specimen surfaces to prevent buckling (Figure 1b). The width of the anti-buckling device is smaller compared to the width in the gauge length area of the specimen (10 mm) to allow for firm gripping on the specimen surface without sliding issues.

The specimens were tested in reference conditions (unstrained material) and after 8% and 12% pre-straining produced by tensile plastic deformation. According to Prosgolitis et al. [25], the reported pre-strain level corresponds to the lower value of plastic strain variation (with 1% tolerance) at the position where extraction of the specimen from the pre-stretched panel under tension took place. That is, the plastic strain of 8% corresponds to a panel location where the plastic strain range was 8–9%, and the value 12% corresponds to a plastic strain range of 12–13% in the initial panel. Fully reversed strain-controlled fatigue tests were performed with amplitudes in the strain range of 0.002–0.015, as shown in Table 2. At least two test repetitions for each combination of parameters were performed. Fracture surfaces were examined using scanning electron microscopy (SEM, JSM-5310, Jeol Ltd., Tokyo, Japan).

## 3. Results and Discussion

### 3.1. Microstructural Features

The microstructural characteristics of the two materials (S355MC, S460MC) were investigated by Prosgolitis et al. [25]. According to the analysis performed, the S460MC steel exhibits a more refined grain structure. In the unstrained material, the grains have an almost equiaxed shape with a slight directionality perpendicular to the rolling direction. Pre-straining induces further deformation of grains in the direction of stretching, which is more pronounced in the case of the S460MC material.

### 3.2. Tensile Properties

The tensile mechanical properties (Table 3) for reference (0% pre-strain), 8%, and 12% pre-strain level, as determined in Prosgolitis et al. [25], are used here for comparison with the cyclic material properties.

The S460MC material exhibits a higher yield and tensile strength but a lower elongation at fracture compared to the S355MC material. The S460MC steel contains Nb, which leads to a finer ferrite grain size after the TMCP (Thermo-Mechanically Controlled Process). This is caused by Zener grain boundary pinning by NbC carbides and the solute drag effect of Nb in solution in austenite during hot rolling. In addition, during accelerated cooling, there is some NbC precipitation in ferrite. The finer grain size and NbC in ferrite explain the higher yield and tensile strength of S460C over S355MC steel.

With increasing pre-strain level, a gradual increase in yield and tensile strength and a decrease in elongation in both steels is observed. At 12% pre-strain, the percentage increase reaches 34.2% and 12.1% for yield and tensile strength, respectively, in the S355MC material. In the S460MC material, the percentage increase is 29.6% and 10.7%, respectively. Elongation is reduced by approximately 40–60% due to pre-straining. Strain hardening is more intense in the S355MC steel due to the more intense dislocation interactions associated with higher dislocation mean free paths.

### 3.3. Strain–Life Diagrams

The strain–life diagrams of the materials have been analyzed in elastic, plastic, and total strain components under both reference and pre-strained conditions, as shown in Figure 2 and Figure 3. The total strain amplitude is given as the sum of the elastic (ε_a,el_) and plastic (ε_a,pl_) part using Basquin and Manson–Coffin relationships, respectively, in the form [27]:(1)$$\varepsilon_{a,t}=\varepsilon_{a,\mathrm{el}}+\varepsilon_{a,\mathrm{pl}}=\frac{\sigma_{f}^{\prime }}{E}{\left(N_{f}\right)}^{b}+\varepsilon_{f}^{\prime }{\left(N_{f}\right)}^{c}\;$$
where E is the Modulus of elasticity, $\sigma_{f}^{\prime }$. is the cyclic stress coefficient, b is the cyclic stress exponent, $\varepsilon_{f}^{\prime }$. is the cyclic ductility coefficient, c is the cyclic ductility exponent, and N_f_ the number of cycles to failure.

The effect of plastic pre-strain level on the materials’ LCF performance is depicted in Figure 2, where the strain–life behavior is displayed in log–log scale using the least squares approximation method on the experimental data points. Pre-straining influences fatigue performance at lives near and below the transition life region. In the S355MC steel, plastic pre-straining improves fatigue performance compared to the unstrained reference material. Although differences in fatigue lives may seem small, the effect of pre-straining is prominent when taking into account the log–log scale in the diagrams. In particular, the percentage increase in fatigue life of the pre-strained material is 18% with 0.002 strain amplitude and about 58% with 0.015 strain amplitude. On the contrary, fatigue lives in the S460MC pre-strained steel are decreased by 17% with 0.012 strain amplitude compared to the reference condition. The difference in the fatigue performance of the two steels is correlated to their cyclic behavior, which is discussed in the next section. Near the transition life, and for strain amplitudes between 0.004 and 0.008, the fatigue life of both steels is not significantly affected by the pre-straining.

The cyclic material constants $\sigma_{f}^{\prime }$, b,
$\varepsilon_{f}^{\prime }$, and c, as evaluated from the strain–life data, are given for each material condition in Table 4.

It is observed that while ductility decreases with pre-straining in both materials (as shown in Table 3), the $\varepsilon_{f}^{\prime }$. parameter decreases only in the S460MC material after pre-straining with regard to the reference condition. In the S355MC steel on the other hand, the $\varepsilon_{f}^{\prime }$ parameter increases with increasing pre-strain level. The observed opposite trend suggests that while the ductility dependence of the $\varepsilon_{f}^{\prime }$ parameter is generally acknowledged, cyclic behavior and its impact on fatigue performance may also influence the value of $\varepsilon_{f}^{\prime }$. The effect observed here may be attributed to the transitional cyclic behavior of the materials under large strain amplitudes, which is discussed in Section 3.4.

In Figure 3, the LCF performance of the S355MC and S460MC steels is compared. The S355MC material exhibits better LCF performance in the examined strain amplitude range. While the S460MC steel has a higher elastic strain part (due to higher yield strength), the better plastic behavior of S355MC renders its overall LCF performance as superior.

An important observation arising from examination of the elastic and plastic components in the strain–life curves is that, in both materials, fatigue transition life decreases with increasing pre-strain level. Specifically, transition fatigue life is reduced in S355MC steel, decreasing from 15,600 cycles in the reference conditions to 9800 cycles in the 12% pre-strained condition; in S460MC steel, the decrease is from 2800 to 2200 cycles, respectively.

### 3.4. Cyclic Behavior

The cyclic stress–strain curves were determined from the stabilized hysteresis loops (Figure 4) and are presented in Figure 5. The stabilized hysteresis behavior is obtained using the cyclic stress–strain values at half the number of cycles of crack initiation life. Crack initiation life is defined as the number of cycles leading to a 10 % drop (when crack initiation occurs within the measured length) or increase (when crack initiation occurs outside the measured length) in the maximum applied load [26].

By comparing the cyclic stress–strain curves with the respective monotonic curves from Figure 5, cyclic softening is the predominant mechanism in the pre-strained materials. This is caused by dynamic recovery processes, mostly cross slip, with which dislocation glide may proceed at lower flow stresses. Pre-staining intensifies this cyclic softening behavior. The decrease in cyclic yield strength (σ_yc_) compared to the monotonic value (σ_y_), due to cyclic softening, reaches values of 31.2% and 33.4% for 8% and 12% pre-straining conditions, respectively, for the S355MC steel. In the S460MC steel, the respective reduction is 28.7% and 29.1%. In reference conditions, the materials exhibit a mixed type of cyclic behavior, including cyclic softening, at strain amplitudes up to 0.008 and 0.012 for the S355MC and S460MC material, respectively, followed by cyclic hardening at higher strain amplitudes.

The values of cyclic yield strength σ_yc_, cyclic hardening exponent n′, and cyclic hardening coefficient K′ as assessed with the Ramberg–Osgood equation (Equation (2)) [28] are presented in Table 5.
(2)$$\varepsilon_{a,t}=\frac{\sigma_{\alpha }}{E}+{\left(\frac{\sigma_{\alpha }}{\;K\;\prime }\right)}^{1/\;n\;\prime }\;$$

In both materials, cyclic yield strength increas with the increases in plastic pre-strain level, whereas the cyclic strain hardening exponent n′ and strength coefficient K′ exhibit opposite trends. For the S355MC material, n′ and K′ decrease with increasing prior plastic deformation, whereas an increase of n′ and K′ is observed in the S460MC material. In the S355MC steel, n′ decreases by 32.7% and K′ by 25.3% at 12% pre-strain, while cyclic yield stress increases by 13.4%. In the S460MC steel, n′ and K′ increase by 8.2% and 8.1%, respectively, and cyclic yield strength increases by 5.9% at 12% pe-strain compared to the reference material.

At 12% pre-strain, the S460MC steel displays the highest cyclic yield strength (450 MPa) and the lowest cyclic strain hardening exponent (0.0955) amongst the materials. The reference S355MC steel exhibits the highest cyclic strain hardening exponent (0.2051) and the lowest cyclic yield strength (300 MPa). The behavior in terms of cyclic properties is consistent with the monotonic behavior of the steels.

The variation in tensile peak stress during cyclic loading is presented in Figure 6, where the cyclic softening behavior of the pre-strained materials is evident. Cyclic softening becomes more prominent with the increase in cyclic strain amplitude, whereas the effect of plastic pre-strain level does not seem to have a significant impact on cyclic material response, leading to similar levels of softening. The reference materials display cyclic softening, which gradually transforms to cyclic hardening with increasing strain amplitude. In the S355MC material, the transition is obtained sooner than in S460MC at a strain amplitude of 0.012, while the S460MC material shows signs of hardening effects at higher cyclic strain amplitudes above 0.0015.

The LCF performance presented in Figure 2 is moderately impacted by pre-straining, while the influencing factors are ductility and cyclic softening/hardening behavior. In the S460MC material, pre-straining degrades ductility and induces cyclic softening, resulting in a decrease in fatigue life with increasing pre-strain level compared to the unstrained condition. S355MC has higher ductility compared to S460MC and superior LCF performance in unstrained conditions. Its fatigue performance seems to suffer in the shift from cyclic softening to cyclic hardening with increasing strain amplitude compared to the pre-strained condition where cyclic softening dominates the behavior. This is attributed to the more intense dislocation interactions due to the larger mean free path and is conceived macroscopically by the low value of cyclic yield strength and high value of n′, factors that explain this dissimilar behavior compared to the other conditions. After pre-straining, the cyclically softened S355MC, despite the lower ductility, results in higher fatigue life. The above results showcase that ductility, as a stand-alone mechanical property, is not sufficient to describe LCF performance and that cyclic softening or hardening play an important role in the material’s behavior.

The above observations are also valid in the comparison of LCF performance between the two materials depicted in Figure 3. In pre-strained materials under cyclic softening conditions, the S355MC steel with higher ductility exhibits better LCF resistance compared to S460MC (Figure 3b,c). In reference conditions, fatigue performance near the transition life is moderately higher in S355MC. With increasing strain amplitude, S355MC shifts from cyclic softening to hardening at lower strain amplitudes than S460MC, which remains under cyclic softening. This dissimilar cyclic behavior elucidates the deterioration in fatigue performance of S355MC compared to the S460MC material (Figure 3a). It should be noted that, based on the definition of crack initiation life in the diagrams of Figure 6, the steeply rising or decreasing stress peaks near the fatigue life are a result of the combined strain-controlled testing condition and crack initiation occurring outside or inside the extensometer gauge length, respectively.

### 3.5. Fractographic Observations

Fractographic analysis was performed in order to reveal the possible effects of pre-straining on low cycle fracture behavior following the onset of fatigue crack initiation. The specimens examined have been subjected to fatigue with 0.002 and 0.015 strain amplitudes. Fracture surfaces were studied in unstrained conditions and after 12% plastic pre-straining. As shown in Figure 7, all surfaces exhibit a fatigue section and a fast fracture section.

The fatigue section has a semi-elliptical shape, which is characteristic of these grades of steel [29] and is smaller in the pre-strained specimens for both steels at all strain amplitudes. The fatigue section in S355MC is smaller by 11.9% (from 6.47 mm^2^ to 5.7 mm^2^) for strain amplitude 0.002 and by 45.3% (from 4.35 mm^2^ to 2.38 mm^2^) for strain amplitude 0.015 (Figure 8) compared to the reference material. Τhe reduction in the fatigue section in the pre-strained S460MC compared to the reference material is 35% (from 5.723 mm^2^ to 3.72 mm^2^) and 32.8% (from 4.85 mm^2^ to 3.26 mm^2^) at strain amplitudes of 0.002 and 0.015 respectively, as shown in Figure 9. Taking into account that a large portion of the fatigue section is included within the low cycle fatigue life determined experimentally after 10% load drop, in the majority of the cases examined, a smaller fatigue section corresponds to a shorter fatigue life [30]. Hence, pre-straining reduces the fatigue resistance of the specimens, an observation that is consistent with the experimental results for the S460MC steel but in contradiction with the results for the S355MC steel. Further investigation was then carried out to obtain an estimate of the crack growth rate using striation spacing measurements.

Fatigue striations were measured in the fatigue section of specimens tested at a strain amplitude of 0.015 with a crack length of 1mm (Figure 7b), and the measurements are given in Table 6. Striations observed on the fracture surfaces are more visible at high strain amplitudes (e.g., 0.015 vs. 0.002). The striation spacing decreases significantly in the case of the pre-strained S355MC steel, while in the S460MC steel the effect is less obvious, showing comparable values for striation spacing in the unstrained and pre-strained material. Although, as discussed in the previous paragraph, the smaller fatigue section in the pre-strained S355MC suggests a shortened fatigue life compared to the unstrained material, the fact that it is accompanied by smaller striation spacing indicates that the crack front advances at a smaller rate within this region [31]. The slower fatigue crack growth rate, which explains the initially observed contradiction, eventually results in larger fatigue life for the pre-strained compared to the reference S355MC steel. Contributing to the short fatigue lives of unstrained S355MC steel at large strain amplitudes is its transition to cyclic hardening behavior (Figure 7a). Pronounced cyclic hardening has also been found to contribute to shortened fatigue lives under large strain amplitudes in TRIP steels [32].

In the present study, the texture effect due to plastic pre-straining [33,34,35], as a further potential influence on the material’s LCF performance, has not been investigated since the research focused on the macroscopic mechanical behavior of the steels, and its potential influence could be considered in a future study.

The fast fracture section displayed in Figure 8c exhibits a dimple formation, typical of ductile fractures for both steels irrespective of pre-strain and strain amplitude.

## 4. Conclusions

The effect of plastic pre-straining on the LCF performance, cyclic properties, and cyclic softening/hardening behavior of the S355MC and S460MC steels was investigated, and SEM fractography was used to analyze fracture characteristics following the onset of fatigue crack initiation. The main conclusions that can be drawn from the conducted research are the following:In unstrained conditions, the S460MC material has higher yield and tensile strength but lower elongation compared to the S355MC material. With increasing plastic pre-strain, yield strength in both materials increases and elongation decreases considerably. Cyclic yield strength increases after pre-straining in both materials, while the cyclic strain hardening exponent increases in S460MC and decreases in S355MC steel with the increase in pre-strain level.The transition life from low to high cycle fatigue decreases with increasing plastic pre-straining in both S355MC and S460MC materials.LCF performance is influenced (apart from ductility) by the cyclic behavior of materials. Plastic pre-straining induces cyclic softening in both steels and, in the S460MC material, LCF performance in the presence of cyclic softening is degraded compared to the unstrained material. The transition from cyclic softening to hardening at large strain amplitudes in the reference S355MC material negatively impacts its LCF performance with regard to the pre-strained conditions.SEM fractography revealed that pre-straining in both materials results in a reduction in the fatigue section with regard to the unstrained material condition. The S355MC steel showed smaller fatigue striation spacing in the pre-strained compared to the unstrained condition, indicative of a lower crack growth rate, which explains the deterioration in fatigue life of the unstrained compared to pre-strained S355MC steel. In the case of S460MC steel, the differences in striation spacing between unstrained and pre-strained material were small.

## Figures and Tables

**Figure 1 materials-15-07927-f001:**
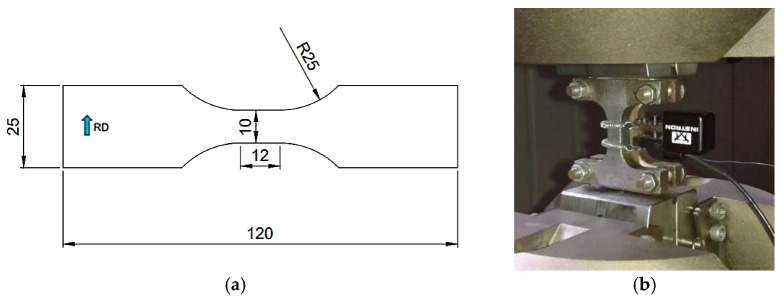
(**a**) LCF test specimen configuration (dimensions in mm); (**b**) strain-controlled fatigue test setup with the use of an anti-buckling device.

**Figure 2 materials-15-07927-f002:**
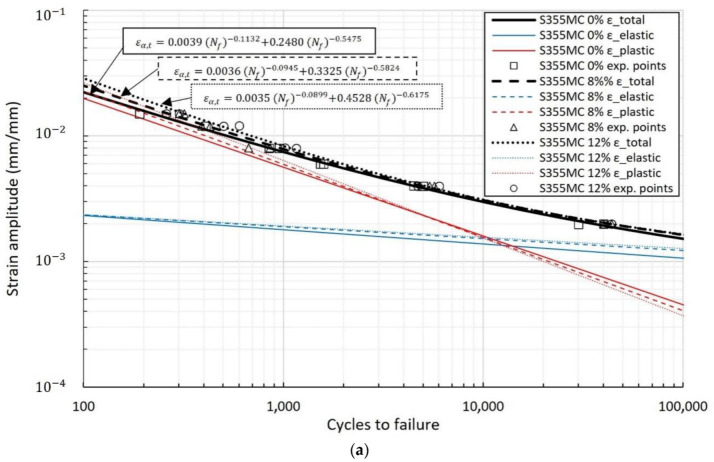
Strain–life curves showing total, elastic, and plastic components for (**a**) the S355MC and (**b**) the S460MC steel.

**Figure 3 materials-15-07927-f003:**
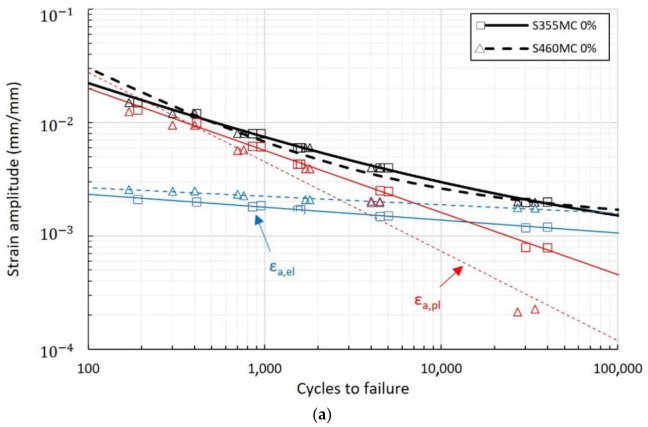
Strain–life curves of the S355MC and S460MC steels for (**a**) reference conditions, (**b**) 8% pre-strain, and (**c**) 12% pre-strain.

**Figure 4 materials-15-07927-f004:**
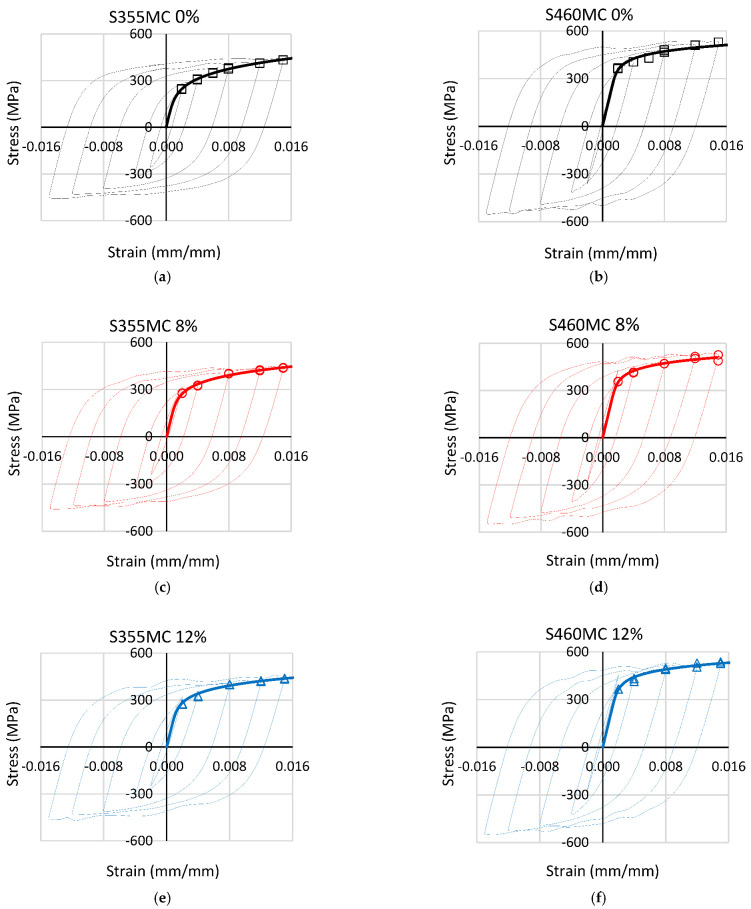
Stabilized stress–strain hysteresis loops of (**a**,**c**,**e**) S355MC and (**b**,**d**,**f**) S460MC steel for different pre-strain conditions.

**Figure 5 materials-15-07927-f005:**
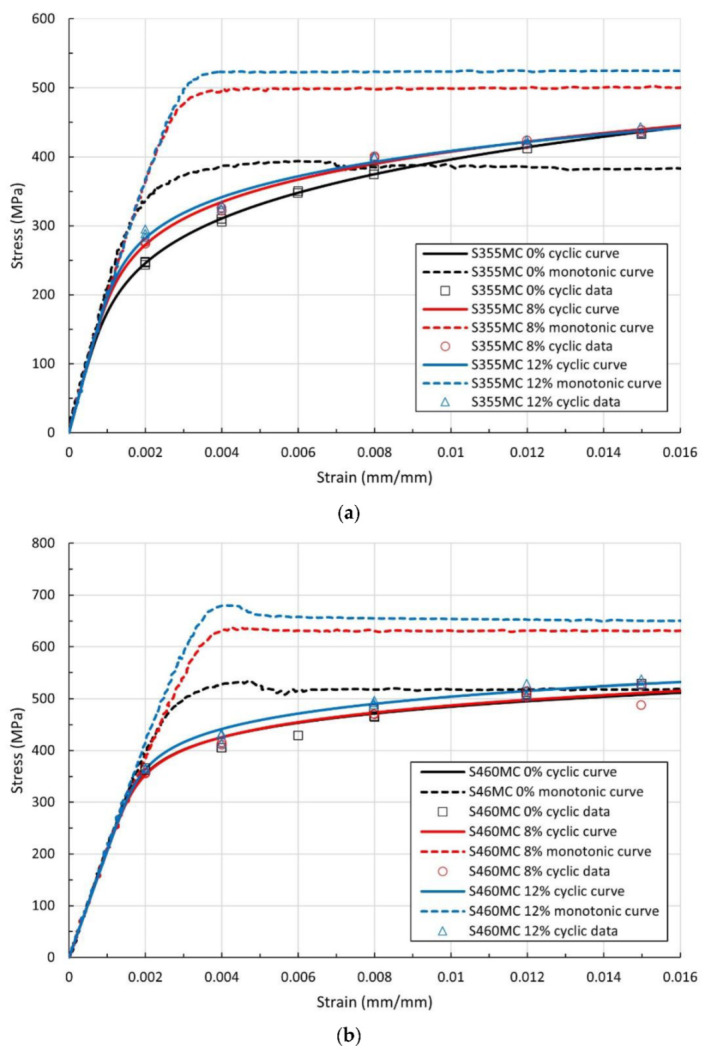
Comparison of monotonic and cyclic σ-ε curves of (**a**) S355MC and (**b**) S460MC steel.

**Figure 6 materials-15-07927-f006:**
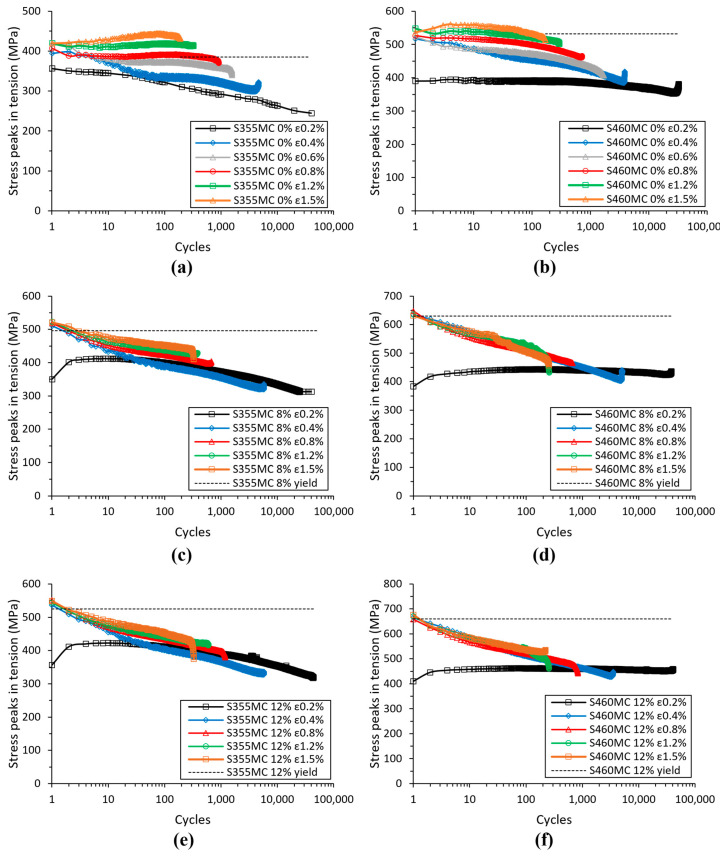
Evolution of cyclic stress peaks under tension with the number of cycles for S355MC steel in (**a**) reference, (**b**) 8% pre-strain, and (**c**) 12% pre-strain material conditions and for S460MC steel in (**d**) reference, (**e**) 8% pre-strain, and (**f**) 12% pre-strain material conditions.

**Figure 7 materials-15-07927-f007:**
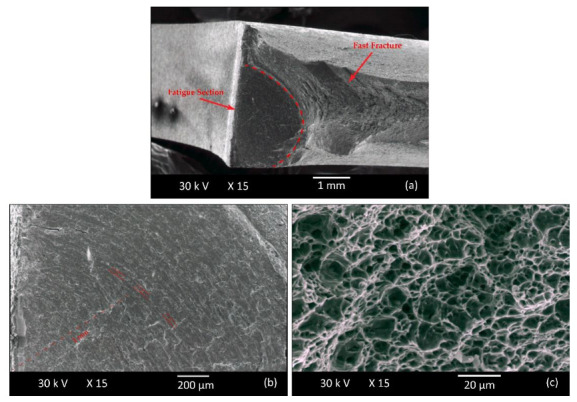
(**a**) Fracture surfaces with (**b**) fatigue and (**c**) fast fracture sections.

**Figure 8 materials-15-07927-f008:**
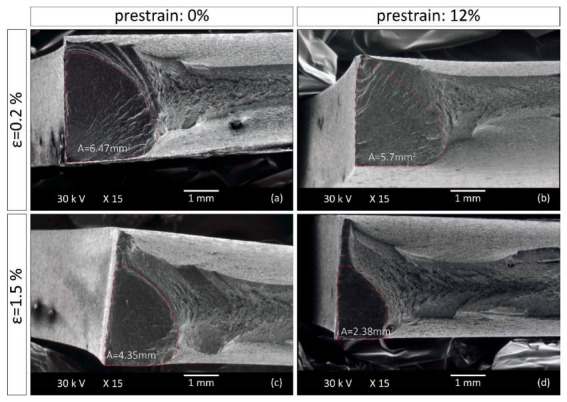
Fatigue section size of fractured S355MC specimens in (**a**) reference and (**b**) 12% pre-strain condition that have been subjected to 0.002 strain amplitude, and in (**c**) reference and (**d**) 12% pre-strain condition that have been subjected to 0.015 strain amplitude.

**Figure 9 materials-15-07927-f009:**
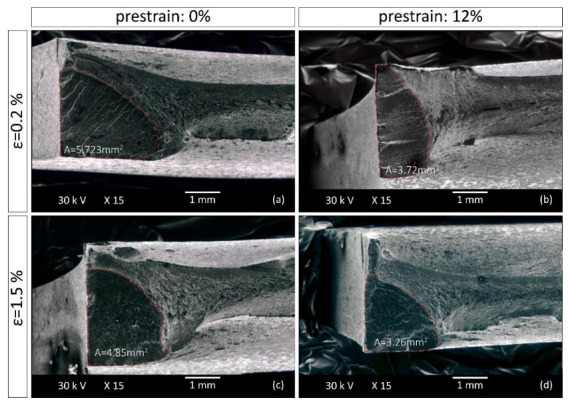
Fatigue section size of fractured S460MC specimens in (**a**) reference and (**b**) 12% pre-strain condition that have been subjected to 0.002 strain amplitude, and in (**c**) reference and (**d**) 12% pre-strain condition that have been subjected to 0.015 strain amplitude.

**Table 1 materials-15-07927-t001:** Chemical composition (wt.%) of S355MC and S460MC steels.

	C	Si	Mn	P	S, Ti, V	Mo	Nb	Cu	Ni	Cr	N	Al
S355MC	0.07	<0.01	0.57	0.01	<0.01	0.01	0.02	0.01	0.04	0.03	0.003	0.03
S460MC	0.06	0.20	0.83	0.03	<0.01	0.01	0.06	0.01	0.02	0.04	0.009	0.03

**Table 2 materials-15-07927-t002:** LCF testing conditions.

Material	Pre-Strain Level (%)	Strain Amplitude
S355MC	0, 8, 12	0.002
		0.004
		0.006
		0.008
		0.012
		0.015
S460MC	0, 8, 12	0.002
		0.004
		0.006
		0.008
		0.012
		0.015

**Table 3 materials-15-07927-t003:** Tensile properties [25].

Material	Plastic Pre-Strain Level (%)	Yield Strength σ_y_ (MPa)	Tensile Strength σ_UTS_ (MPa)	Elongation at Fracture (%)
S355MC	0	380	470	41
	8	480	512	23
	12	510	527	20
S460MC	0	490	590	27
	8	610	630	16
	12	635	653	11

**Table 4 materials-15-07927-t004:** Cyclic material parameters.

Material	Pre-Strain Level	$\sigma_{f}^{\prime }\;(\mathrm{MPa})$	b	$\varepsilon_{f}^{\prime }\;$	c
S355MC	0%	804.94	−0.1132	0.2480	−0.5475
	8%	746.29	−0.0945	0.3325	−0.5824
	12%	735.42	−0.0899	0.4528	−0.6175
S460MC	0%	773.06	−0.0748	1.0473	−0.7887
	8%	748.19	−0.0702	0.6057	−0.7150
	12%	782.59	−0.0726	0.8427	−0.7696

**Table 5 materials-15-07927-t005:** Cyclic properties.

Material	Pre-Strain Level	n′	K′ (MPa)	σ_yc_ (MPa)
S355MC	0%	0.2051	1064.60	300
	8%	0.1608	885.18	330
	12%	0.1381	795.52	340
S460MC	0%	0.0883	740.04	425
	8%	0.0960	775.90	435
	12%	0.0955	799.71	450

**Table 6 materials-15-07927-t006:** Measurement of fatigue striations.

Material	Pre-Strain Level	Striation Spacing (μm)
S355MC	0%	23 ± 2
	12%	11 ± 2
S460MC	0%	18 ± 1
	12%	15 ± 1

## Data Availability

The data presented in this study are available on request from the corresponding author. The data are not publicly available due to forming part of an ongoing study.

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
