# Peer review of "Low Cycle Fatigue Behavior of Plastically Pre-Strained HSLA S355MC and S460MC Steels"

_materials, 2022, doi:10.3390/ma15227927_

Round 1

Reviewer 1 Report

 The study is devoted to the actual problem of the effect of preliminary deformation on the mechanical properties of two steels of different strength and their behavior under conditions of subsequent cyclic loading. The work is well thought out, includes the evaluation of the strain hardening coefficients and the quantitative analysis of the macro- and microrelief of fracture. The obtained data are important from a practical and scientific point of view, the paper is recommended for publication.

Author Response

The authors would like to thank the reviewer for his comments on the manuscript.

Reviewer 2 Report

The article has a scientific character. The article deals with low-cycle fatigue problem of two structural steels. The Authors applied correct research methods and used the appropriate measuring equipment. The content of the work is logically written. The manuscript contains 9 figures and 6 tables. Figures and tables are properly prepared. Authors cited 33 literature sources. The authors presented an interesting work and pointed to the novelty aspect. The work requires only careful editing of the text.

I will recommend publishing the article.

Author Response

The authors would like to thank the reviewer for the comments on the manuscript and have improved the text based on his suggestion

Reviewer 3 Report

This paper investigated the LCF performance of plastically prestrained S355MC and S460MC steels. The materials were subjected to plastic pre-stretching prior to cyclic testing, and the obtained LCF behavior was tested and discussed under different levels of plastic pre-straining and compared with the behavior of initially unstrained materials. The experimental fatigue lives were combined with cyclic hardening or softening characteristics of the materials and SEM fractography to further investigate fracture characteristics following onset of fatigue crack initiation.

The paper is well-structured and well-discussed. The findings in this paper are interesting even though they are just a small step on long way of knowledge acquisition on the LCF behavior of metal materials. In the opinion of the reviewer, only minor revisions and explanations are needed before it can be accepted for publication in the journal.

1. The fatigue section and the fast fracture section are suggested to be marked respectively in Fig. 7.

2. When the anti-buckling device was used, how can the axial clip-on extensometer be attached firmly on the specimen surface without any slides?

3. As 6 different strain amplitudes were adopted during the LCF tests, why not 6 points are observed in all the subfigures of Fig. 4?

4. For the evolution of cyclic stress peaks in tension with some certain number of cycles shown in Fig. 6, why a steep rise can be observed near the fatigue life, e.g., S460MC 8%-ε0.4% and S355MC 0%-ε0.4%?

Author Response

Response to Reviewers Comments

Point 1. The fatigue section and the fast fracture section are suggested to be marked respectively in Fig. 7.

Response 1: In Fig. 7 markings in red colour have been used to distinguish the fatigue and fast fracture section according to the Reviewer’s comment.

 Point 2. When the anti-buckling device was used, how can the axial clip-on extensometer be attached firmly on the specimen surface without any slides?

 Response 2: The width of the anti-buckling device is smaller compared to the LCF specimen. The axial clip-on extensometer adjusts to the specimen’s side surface using two springs to keep it in tight contact with the specimen (as shown in Figure 1b in the manuscript). The explanation has been included in the marked up manuscript in lines 109-111.

Point 3. As 6 different strain amplitudes were adopted during the LCF tests, why not 6 points are observed in all the subfigures of Fig. 4?

Response 3: Strain amplitude 0.006 has been examined only in the unstrained materials. For this reason, six points are observed only in subfigures 4(a) and (b), while in subfigures 4(c),(d),(e),(f) five points corresponding to strain amplitudes 0.002, 0.004, 0.008, 0.012, 0.015 are presented.

 Point 4. For the evolution of cyclic stress peaks in tension with some certain number of cycles shown in Fig. 6, why a steep rise can be observed near the fatigue life, e.g., S460MC 8%-ε0.4% and S355MC 0%-ε0.4%?

Response 4: As stated on page 9, lines 207-209 of the manuscript, “Crack initiation life is defined as the number of cycles leading to a 10 % drop (when crack initiation occurs within the measured length) or increase (when crack initiation occurs outside the measured length) of the maximum applied load”. Thus, the steep rise of stress near the fatigue life means that crack initiation occurs outside the extensometer gauge length.The clarification has been included in lines 273-276 in the revised manuscript.

Reviewer 4 Report

The article's subject is current and concerns low-cycle fatigue tests of plastically pre-strained steels, which justifies the article's publication. The paper is structured correctly and contains well-described charts and photos. Still, the cited literature only includes references to a few items from recent years that need to be supplemented. In general, the interpretation of the results and illustrations is correct, and the presented conclusions well capture the obtained research results.

The work concerns the engineering problem of the influence of pre-strain on the mechanical and fatigue properties of two different strength steel. The work's strength is analysing the materials' behaviour under variable load controlled by strain, including evaluating strain hardening factors and macro- and microfractographic analysis. The compiled data constitute a coherent study that may arouse interest from a scientific and practical perspective. The paper is recommended for publication.

Proposed amendments:

1.       The state of art can be enriched with literature items from the recent time.

2.       Figure 2, please check the total strain amplitude indications in the picture.

Author Response

Response to Reviewers Comments

Point 1. The state of art can be enriched with literature items from the recent time.

Response 1: The state of the art has been further updated by including two more cited references from the years 2019 and 2020 (new references 21 and 22), which are analysed in lines 74-81 in the revised manuscript.

Point 2. Figure 2, please check the total strain amplitude indications in the picture.

Response 2: The total stain amplitude in the Figure has been checked for correctness according to the Reviewer’s suggestion.
